

# Space-time mapping relationships in sensorimotor communication during asymmetric joint action

Ke Zhang[1], Xin Tong[1], Shaofeng Yang[1,2,3], Ying Hu[1,2], Qihan Zhang[1,2] and Xuejun Bai[1,2]

[1] Department of Psychology, Tianjin Normal University, Tianjin, China
[2] Academy of Psychology and Behavior, Tianjin Normal University, Tianjin, China
[3] School of Psychology, Inner Mongolia Normal University, Hohhot, China

Corresponding authors
Qihan Zhang, zqhouou@126.com
Xuejun Bai, bxuejun@126.com

## ABSTRACT

**Background.** Sensorimotor communication is frequently observed in complex joint actions and social interactions. However, it remains challenging to explore the cognitive foundations behind sensorimotor communication.

**Methods.** The present study extends previous research by introducing a single-person baseline condition and formulates two distinct categories of asymmetric joint action tasks: distance tasks and orientation tasks. This research investigates the action performance of 65 participants under various experimental conditions utilizing a 2 (cooperative intention: Coop, No-coop) × 2 (task characteristic: distance, orientation) × 4 (target: T1, T2, T3, T4) repeated-measures experimental design to investigate the cognitive mechanisms underlying sensorimotor communication between individuals.

**Results.** The results showed that (1) target key dwell time, motion time, total motion time, and maximum motion height in the Coop condition are more than in the No-coop condition. (2) In the distance task without cooperative intention, the dwell time of T4 is smaller than T1, T2, T3, and its variability of T1, T2, T3, and T4 were no different. In the distance task with cooperative intention, the dwell time and its variability of T1, T2, T3, and T4 displayed an increasing trend. (3) In the orientation task without cooperative intention, the dwell time of T1 is smaller than T2, T3, T4, and variability of the target keys T1, T2, T3, and T4 had no difference. In the orientation task with cooperative intention, the dwell time and variability of the target keys T1, T2, T3, and T4 had increasing trends.

**Conclusions.** Those findings underscore the importance of cooperative intention for sensorimotor communication. In the distance task with cooperative intention, message senders establish a mapping relationship characterized by "near-small, far-large" between the task distance and the individual's action characteristics through sensorimotor experience. In the orientation task with cooperative intention, message senders combined sensorimotor experience and verbal metaphors to establish a mapping relationship between task orientation and action characteristics, following the sequence of "left-up, right-up, left-down, right-down" to transmit the message to others.

## INTRODUCTION

As members of social groups, humans can inherently engage in social interactions. Before infants acquire language skills, they demonstrate the capacity to communicate and interact with others through nonverbal actions (*Oryadi-Zanjani, 2020*). Nonverbal communication permeates human cultures worldwide, often complementing or replacing verbal communication in everyday social interactions (*Hall, Horgan & Murphy, 2019*). Abundant evidence suggests that individuals frequently employ nonverbal sensorimotor communication to swiftly convey coordination signals in the context of real-time social interactions or joint actions (*Laroche et al., 2022*; *Miyata et al., 2021*; *Edey et al., 2020*; *Varni et al., 2019*). In other words, individuals convey information to others by embedding communicative messages within instrumental actions (*Pezzulo et al., 2019*) to facilitate the coordination of interindividual interactions, a phenomenon referred to as sensorimotor communication (SMC). For instance, in competitive sports, an athlete may intentionally modify his kicking to convey the upcoming coordination direction to teammates. Here, the initial kicking action serves as an instrumental act, while the information regarding the coordination direction (manifested as an exaggerated deviation in the individual's kicking trajectory) is communicative. Likewise, athletes can execute deceptive body movements that disrupt their opponents' motor prediction processes. Sensorimotor communication relies on instrumental actions and enables the conveyance of communicative information during the execution of instrumental actions. Information transfer in sensorimotor communication is highly flexible and rapid (*Laroche et al., 2022*; *Vesper, Schmitz & Knoblich, 2017*). Swift information transfer between message senders and receivers through actions is achievable even without prior agreement among interacting parties regarding the meaning of the action (*Pezzulo et al., 2019*). Consequently, it is frequently observed in complex joint actions and social interactions.

Asymmetric joint action is a relatively complex type of joint actions because it necessitates spatial and temporal coordination among participants who receive incongruent information (*Zhang, 2019*; *Vesper, Schmitz & Knoblich, 2017*). For instance, two individuals are instructed to touch a designated target location sequentially. One of them possesses knowledge of the target location, while the other remains unaware. Sensorimotor communication plays an essential role in joint action because effective motor coordination can only be achieved if the participant who possesses more information (the information sender) conveys the target information to the less informed participant (the information receiver). The bidirectional model of influence asserts that effective communication hinges on the sender's precise articulation of the message to ensure comprehension by the receiver (*Beebe, Beebe & Ivy, 2015*). Consequently, the precise calibration of the kinematic characteristics of action by message senders, such as motion height, motion time, and motion speed (*Trujillo, 2020*) based on communicative information, is a prerequisite for sensorimotor communication to enable asymmetric joint action. The process by which message senders establish the mapping between task target information and their action characteristics assumes particular significance.

Previous research in the domain of asymmetric joint action has established that message senders possess the capability to adjust the kinematic characteristics of their actions in correspondence with changes in the physical attributes of the task target. For instance, *Schmitz et al. (2018)* observed that message senders effectively conveyed three different weight categories—light, medium, and heavy—by grasping a cylinder at varying heights. Specifically, they grasped it at a higher position to indicate a lighter weight, a middle position for a medium weight, and a lower position for a heavy weight. Furthermore, *Vesper, Schmitz & Knoblich (2017)* noted that message senders adapted their motion time based on the distance to the task target, with longer motion times represented more distant targets. These observations align with the theory of embodied cognition, which underscores the profound influence of bodily actions and sensory experiences on forming abstract concepts (*Ye, 2010*; *Li & Wang, 2015*). Sensorimotor experiences are bodily actions and sensory experiences (*Jin et al., 2019*; *Ye, 2010*). According to this theory, when individuals engage with concepts, relevant embodied simulations, and neural systems are activated even when there is no real-time, online interaction with these concepts (*Barsalou et al., 2008*; *Barsalou, 2009*). In sensorimotor communication, processing the weight/motor distance information of a task target automatically activates the corresponding sensorimotor experiences, subsequently influencing the grasp height/motion time of message senders' actions.

However, the studies mentioned above leave specific critical questions unanswered. First, although these investigations confirm that message senders adapt the kinematic characteristics of their actions based on the target, none of them compare these actions with the kinematic characteristics of actions performed by individuals in the tasks without cooperation. Consequently, it remains challenging to discern whether the disparities in message senders' actions stem from variances in instrumental movements associated with distinct task targets. Alternatively, it could be intentional sensorimotor communication by the individuals involved. For instance, in a study by *Schmitz et al. (2018)*, the act of grasping the cylinder by message senders served both an instrumental purpose and a communicative intention. Consequently, the issue of whether the alteration in grasping height results from differences in the object's weight or intentional communicative messages conveyed by the message senders remains elusive. Second, the physical attributes of the targets in the aforementioned research tasks evoke substantial divergence in individual sensorimotor experiences, such as incremental differences in weight (light, medium, and heavy) and incremental changes in distance (near, medium, and far). In such cases, the message senders can readily determine the kinematic characteristics of the corresponding motion by observing variations in the target's physical attributes. However, in intricate social interactions characterized by limited differentiation in target-induced sensorimotor experiences, the way message senders engage in sensorimotor communication warrants exploration.

To address Problem 1, the current study extends prior research by introducing a single-person baseline condition. This addition aims to isolate instrumental action distinctions stemming from task-related factors from the sensorimotor communication of message senders. Additionally, previous investigations have revealed that sensorimotor communication does not manifest uniformly across all phases of message senders'

actions (*Vesper, Schmitz & Knoblich, 2017*). Building upon this insight, the present study deconstructs the action process of message senders. Research has demonstrated that message senders systematically adjust kinematic characteristics (*Trujillo, 2020*; *De Ruiter et al., 2010*) and enhance the informativeness of their actions (*Winner et al., 2019*) contingent on the communicative context to facilitate effective message delivery. This is exemplified by the elongation of motion time (*Vesper et al., 2016*) or an increase in motion amplitude (*Wood et al., 2022*; *McEllin, Knoblich & Sebanz, 2018*). The present study's Hypothesis 1 asserts that message senders tend to amplify specific motion characteristics during particular motion phases when demonstrating cooperative intention (Coop), as compared to a baseline condition when there is no cooperative intention (single-person baseline, No-coop).

To address Problem 2, this study devises two distinct types of asymmetric joint action tasks: distance and orientation tasks. Both task types consist of four target keys, requiring both participants to sequentially press a designated target key. However, only one of the participants possesses knowledge of the target key's location. In the distance task, the targets are placed evenly along the same direction but differ in distance. Conversely, the targets are placed in different directions but cover the same distance in the orientation task. In both task types, message senders are tasked with establishing a mapping relationship between the spatial-physical characteristics of the target key (motion direction and motion distance) and the kinematic attributes of their actions (*e.g.*, motion time). This mapping relationship, known as space–time mapping, conveys the target message and subsequently facilitates joint actions. Specifically, the distance task primarily focuses on the space–time mapping between motion distance (target) and motion time (action). In contrast, the orientation task places greater emphasis on the space–time mapping between motion direction (target) and motion time (action).

In accordance with the theory related to embodied simulation, it is well established that as one moves further away, the accompanying motion time tends to increase (*Sevdalis & Keller, 2011*). Consequently, in a distance task characterized by a more pronounced differentiation in target-induced sensorimotor experiences, message senders can establish space–time mapping relationships between motion distance and motion time through embodied simulations of the spatial distance characteristics of the task target. Therefore, Hypothesis 2a in this study posits that in the distance task with cooperative intention, message senders will extend their motion time in direct proportion to the spatial distance information of the target to effectively convey the target message to others. In the orientation task, the mapping relationships between spatial orientation and time are notably intricate. Forming space–time mappings in orientation tasks solely through target-induced sensorimotor experiences presents considerable challenges, rendering orientation tasks less differentiated. A correlational study examining the Space-Time Association of Response Codes Effect (STARC) identified three primary spatial orientations (left–right, front–back, and up-down) within the mental timeline (*He et al., 2020*; *Coull, Johnson & Droit-Volet, 2018*; *Teghil, Marc & Boccia, 2021*; *Von Sobbe et al., 2019*; *Starr & Srinivasan, 2021*; *Valenzuela et al., 2020*). Due to the influence of reading and writing conventions, the left direction typically represents earlier times, while the right signifies later times

(*Dalmaso, Schnapper & Vicovaro, 2023*; *Pitt & Casasanto, 2020*). This low-level embodied simulation establishes a mental timeline oriented from left to right. In contrast, the mental timeline associated with up-and-down orientation is primarily linked to high levels of verbal metaphors (*He et al., 2021*). For instance, Chinese linguistic metaphors such as "morning (上午)" and "afternoon (下午)" equate to earlier and later times, respectively. "上" represents the up part of the orientation, and "下" represents the down side of the orientation. These linguistic metaphors activate spatial schemas that offer reference points for time processing (*Boroditsky, Fuhrman & McCormick, 2011*). In the present study, the target keys within the orientation tasks encompass four distinct orientations: left-up, right-up, left-down, and right-down. This setup may engage embodied simulation for left–right orientation and utilize linguistic metaphors for up-down orientation. Since embodied simulation rooted in reading and writing habits occurs more frequently than verbal metaphors, producing mental timelines in the "left–right" direction is likely to be more effortless and rapid than those in the "up-down" direction (*Chen, 2018*). Additionally, prior research has indicated that Mandarin-speaking individuals tend to construct their timelines from left-up to right-down (*Hartmann et al., 2014*; *Sun et al., 2022*). Consequently, Hypothesis 2b in this study posits that in the orientation task with cooperative intention, message senders may extend the motion time in correspondence with the target's left-up, right-up, left-down, and right-down orientation sequence to convey the target message to others effectively.

## MATERIALS & METHODS

### Participants

MorePower 6.0.4 was used to calculate the sample size. A sample of at least 60 is required for a 0.8 probability to correctly reject the null hypothesis (power = 0.8) given a medium effect size (two-tailed test, partial $\eta^2 = 0.06$) for the 2 × 2 ×4 within-interaction. A total of 65 participants (36 males, $M_{age} = 20.06$ years, $SD_{age} = 2.80$ years) were recruited from Tianjin Normal University. To control for individual differences, such as arm length and height, which could potentially impact the kinematic indices of participants' arm motion time and height, these attributes were equated before the experiment ($M_{arm} = 68.22$ cm, $SD_{arm} = 4.87$ cm; $M_{height} = 169.66$ cm, $SD_{height} = 9.42$ cm). All participants were right-handed as determined by the Edinburgh Handedness Inventory (*Oldfield, 1971*) and reported normal or corrected-to-normal vision and normal hearing. All participants spoke Mandarin. The participants signed prior informed consent before the experiment and received monetary compensation. The experimental protocol was approved by the ethics committee of Tianjin Normal University (No. 2021030809).

### Experimental design

This study employed a 2 × 2 × 4 within-subjects experimental design, incorporating the factors of cooperative intention (Coop *vs.* No-coop), task characteristic (distance *vs.* orientation), and target (T1, T2, T3, *vs.* T4). The dependent variables encompassed participants' keystroke responses and motion trajectory characteristics in each experimental condition, as elaborated upon in the Data Analysis section.

## Apparatus

The experimental program was developed, and the stimulus presentation was executed using Psychtoolbox 3.0 for MATLAB 2019a (*The MathWorks, Inc, 2019*). The experimental stimuli were displayed on a Dell screen (Model U2417H, 24 inches in size, with a resolution of 1,920 × 1,080 pixels).

Two sets of customized keyboards were employed as response devices, each consisting of five keys with a base size of 3 cm × 3 cm. These keys were connected to transmission lines (each 1 m in length) and were ultimately assembled on a motherboard to create a set of keyboards. Notably, each key on this keyboard could move freely.

For motion tracking, a Nokov optical 3D motion capture system (Mars 4H, NoKov Corporation, Beijing, China), manufactured by Beijing Nokov Science & Technology, was employed. A motion capture marker was affixed to the tip of the participant's right index finger, and seven high-power HLED luminaires (sampling rate = 100 Hz) were used to capture the motion trajectory of the fingertip (marker).

## Experimental setup

The participant was seated in the middle of the table (60 cm in length, 80 cm in width, and 78 cm in height), and the screen was positioned 65 cm away from the participant. A customized keyboard was placed on the table with two types available: the distance keyboard and the orientation keyboard. On the distance keyboard, the starting key was situated five cm from the table's edge, and the intervals between T1, T2, T3, T4 and the starting key were 10 cm, 20 cm, 30 cm, and 40 cm, respectively. The diameters of the keycaps for the starting key, T1, T2, T3, and T4 were two cm, one cm, two cm, three cm, and four cm, respectively. On the orientation keyboard, the starting key was positioned 25 cm away from the table's edge, with consistent 20 cm intervals between T1, T2, T3, T4, and the starting key. All keycaps had a diameter of two cm. Please refer to Fig. 1 for a visual representation of the setup. This configuration was designed to ensure that regardless of the keyboard type, the coefficient of difficulty calculated by Fitts' law (Eq. (1); *Fitts, 1954*; *Vesper, Schmitz & Knoblich, 2017*) for a participant moving from the starting key to each target key remained consistent at 4.32. Fitts' law evaluates the relationship between the coefficient of difficulty of the motion (ID) and the amplitude of the motion (A), target width (W).

$$\text{ID} = \log 2 \frac{2A}{w}. \tag{1}$$

## Tasks
### Distance task

The distance task consisted of two variations, with and without cooperative intention. Both employed the distance keyboard.

In the distance task without cooperative intention, participants were tasked with completing a keystroke assignment based on the target cue presented on the screen by responding at a natural pace. This condition served as a baseline for participants' actions under various task targets. Participants were instructed to position the tip of their right
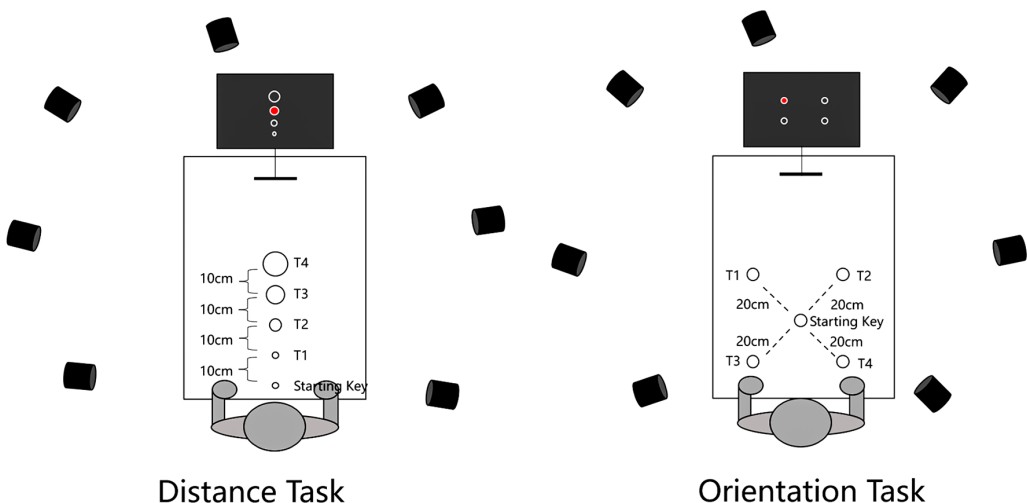

**Figure 1** Schematic illustration of the experimental setup for distance task (left), and for orientation task (right).

index finger at the center of the starting key (starting posture) before each trial. At the beginning of each trial, a target key cue was presented in the center of the screen (2 s) with one of the four target keys highlighted in red. The red dot indicated the target. Following the target key cue presentation, a yellow "+" appeared on the screen, accompanied by a brief "bee" tone (200 ms) to signal the impending task initiation. When the yellow "+" and the "bee" sound vanished, the participants commenced the keystroke task. During the task, a white "+" was displayed on the screen, and participants were required to press the starting key followed by the designated target key (T1/T2/T3/T4). Pressing the starting key triggered a "da" sound, while pressing the target key (T1/T2/T3/T4) resulted in a "di" sound. The dwell time of the "da" and "di" sounds was determined by the dwell time of the participant key press, as depicted in Fig. 2A. Subsequently, participants were instructed to return their fingers to the starting position. A total of 60 trials were conducted for this task, with 15 trials for each target (T1/T2/T3/T4). These 60 trials were randomly divided into four blocks, with randomized orders and intervals ranging from 16 to 24 s between blocks.

In the distance task with cooperative intention, each pair of participants collaborated to complete the task, with Participant A and Participant B working together to press the same target key. During each trial, only Participant A possessed knowledge of the target key's location; Participant B did not. Participant A was required to nonverbally convey the target key's location to Participant B during the key press. Subject B could hear the sound produced by the keys but could not observe the action. In this scenario, Participant A was real and Participant B was virtual. Participant A was told that Participant B was a stranger and a same-sex peer. To enhance the realism of the virtual participants, Participant A was informed before the experiment that Participant B was located in an adjacent lab. Additionally, the experimenter temporarily left the lab for 1–3 min before the experiment began and informed Participant A that she was checking on the readiness of the other

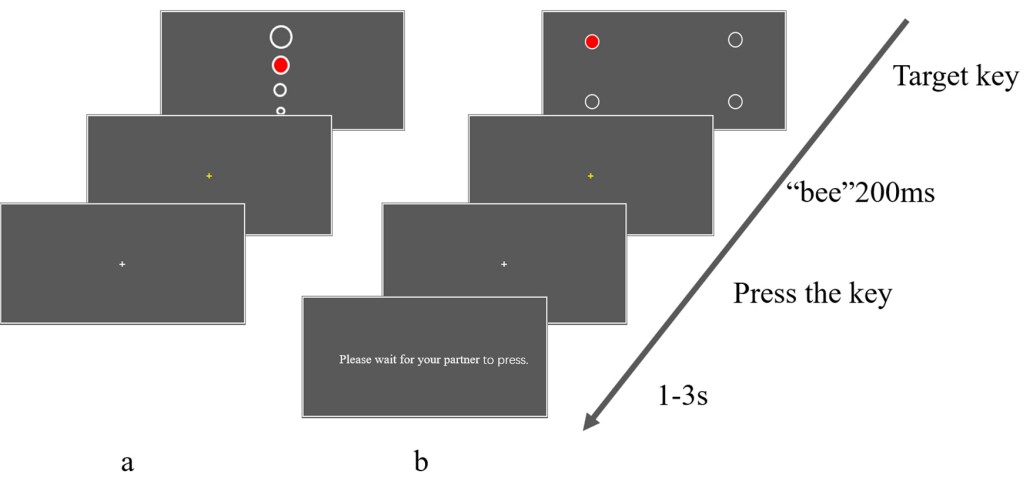

**Figure 2** Flowchart of distance task without cooperative intention (A) and orientation task with cooperative intention (B).

lab. The primary distinction in the distance task with cooperative intention, compared to without cooperative intention, was the appearance of a prompt on the screen that read "Please wait for your partner to press the key" (1–3 s). This prompt was displayed after Participant A completed the motion and returned his or her finger to the starting position. Participant A was informed that his or her partner would press the key during this time. This prompt was introduced to enhance the realism of the virtual participants.

### *Orientation task*

The orientation tasks included two types, with and without cooperative intention. The tasks were the same as the distance task with and without cooperative intention, except for the use of the orientation keyboard. The flow of the orientation task with cooperative intention can be seen in Fig. 2B.

### Procedure

In the preparation phase, participants filled out the informed consent form, personal information form, and Edinburgh Handedness Inventory. The experimenter measured and recorded the participants' height and arm length and then affixed a motion capture marker to the fingertip of their right index finger.

Before the main experiment started, the participants completed four practice trials, one for each target key, to familiarize themselves with the procedure in the No-coop condition. To ensure that the participants fully understood the requirements, the formal experiment proceeded only after the participants successfully executed all four practice trials. During the formal experiment, the participants performed the task first in the No-coop condition and then in the Coop condition. This sequence was designed to prevent the Coop condition from influencing the motion performance of the No-coop condition. The sequences of the distance task and orientation task were counterbalanced between participants.

After the end of the experiment, the participants filled out a questionnaire in which they were asked to explain how they solved the task. The questionnaire consisted of four questions: (1) What kind of person do you think your partner is? (2) Under the distance task with cooperative intention, how did you convey information to your partner, and what strategy did you use? (3) Under the orientation task with cooperative intention, how did you convey information to each other, and what strategies did you use? (4) Did you experience any discomfort or confusion during the entire experiment?

## Data analysis
### Keystroke response

The key press responses of the participants were measured in this study to assess action characteristics at different stages and to evaluate overall action performance. The participants' dwell time (DT) on the target key, which represented the time from pressing to lifting the target key, and the participants' motion time (MT), representing the time from lifting the starting key to pressing the target key, served as metrics for assessing localized action characteristics. The participants' total motion time (TMT), representing the duration from pressing the starting key to lifting the target key, served as an assessment index for holistic action characteristics. After excluding invalid trials, each of the three indicators underwent a 2 (task characteristic: distance, orientation) $\times$ 2 (cooperative intention: Coop, No-coop) $\times$ 4 (target: T1, T2, T3, T4) repeated-measures ANOVA with Bonferroni correction and paired $t$-tests using SPSS (v.23.0; SPSS Inc., Chicago, IL, USA). A statistical threshold of $p < 0.05$ was considered significant.

To assess the quality of sensorimotor communication by message senders, this study also calculated the signal-to-noise ratio ($SNR_{MT}$, $SNR_{DT}$, $SNR_{TMT}$) for the quality of message communication based on the participants' keystroke responses, as outlined in Eq. (2).

$$SNR = \frac{M((MT2-MT1),(MT3-MT2),(MT4-MT3))}{M(SDT1, SDT2, SDT3, SDT4)}. \tag{2}$$

MT1, MT2, MT3, and MT4 represent the average MT/DT/TMT of target keys T1, T2, T3, and T4, and SDT1, SDT2, SDT3, and SDT4 denote the MT/DT/TMT variability of target keys T1, T2, T3, and T4. The signal-to-noise ratios ($SNR_{MT}$, $SNR_{DT}$, $SNR_{TMT}$) for the quality of message communication in the MT, DT, and TMT under different experimental conditions were subjected to 2 (cooperative intention: Coop, No-coop) $\times$ 2 (task characteristic: distance, orientation) repeated-measures ANOVA with Bonferroni correction and paired $t$-tests using SPSS (v.23.0). A statistical threshold of $p < 0.05$ was considered significant. According to the previous hypothesis, it was observed that the space–time mapping relationships between the target key and the motion time under both the distance task and the orientation task with cooperative intention were prolonged in equal proportions with the changes in T1, T2, T3, and T4. Therefore, a larger SNR indicated that the way of communicating information was more aligned with the research hypotheses, resulting in improved quality of the message communication (*Vesper, Schmitz & Knoblich, 2017*).

*Motion trajectory.* It has been established in prior studies that sensorimotor communication by message senders not only alters motion time but may also adjust the
maximum motion (*Candidi et al., 2015*). To comprehensively examine the sensorimotor communication of message senders, this study processed and analyzed motion capture data. Initially, trials featuring incorrect key presses and those lacking recorded motion capture markers were excluded. Subsequently, the motion capture data were preprocessed using Cortex 7.0 software to obtain the motion trajectory of the motion capture marker under each experimental condition, represented as 3D coordinates. Next, a self-programmed script in MATLAB (2019a) was employed to calculate the maximum motion height ($MAX_{MH}$) between the participants' starting key press and the target key lift for each experimental condition. Finally, a 2 (task characteristic: distance, orientation) $\times$ 2 (cooperative intention: Coop, No-coop) $\times$ 4 (target: T1, T2, T3, T4) repeated-measures ANOVA with Bonferroni correction and paired *t*-tests were conducted using SPSS (v.23.0). A statistical threshold of $p < 0.05$ was considered significant.

### Questionnaire

The strategies within the distance task with cooperative intention and the orientation task with cooperative intention in the questionnaire were categorized. Furthermore, a data-driven approach was used to cluster analyze the SNR of the most effective indicators in the orientation task with cooperative intention. This was done to investigate whether participants established a space–time mapping relationship between task targets and participant actions at the subjective level of consciousness.

Furthermore, to ensure the reliability of the statistical results, this study conducted Bayesian repeated-measures ANOVA (*Wang et al., 2023*) on the aforementioned indicators using JASP (0.17), as outlined in the Supplemental Information. The results of the two statistical analyses mentioned above were found to be relatively consistent.

## RESULTS

### Data preparation

Trials that did not align with the experimental requirements were excluded, encompassing two specific criteria: (1) trials in which the key was not pressed in accordance with the target information presented on the screen, and (2) trials in which the target key was pressed before the starting tone ("bee") appeared. Invalid data, amounting to 0.25% of the total, were discarded. Furthermore, data that adhered to the experimental requirements but fell beyond the range of $\pm 3$ standard deviations from the mean of the conditions were categorized as extreme data. These extreme data points, which ranged from 0% to 2.88% for each indicator, were replaced with the mean value.

### Keystroke responses
#### *Whole indicator analysis*

A 2 (task characteristic: distance, orientation) $\times$ 2 (cooperative intention: Coop, No-coop) $\times$ 4 (target: T1, T2, T3, T4) repeated-measures ANOVA was conducted on both holistic and localized indicators. Significantly, the third-order interaction of task characteristic, cooperative intention, and the target was observed solely in total motion time and target key dwell time, as illustrated in Fig. 3. This implied that both total motion time and target

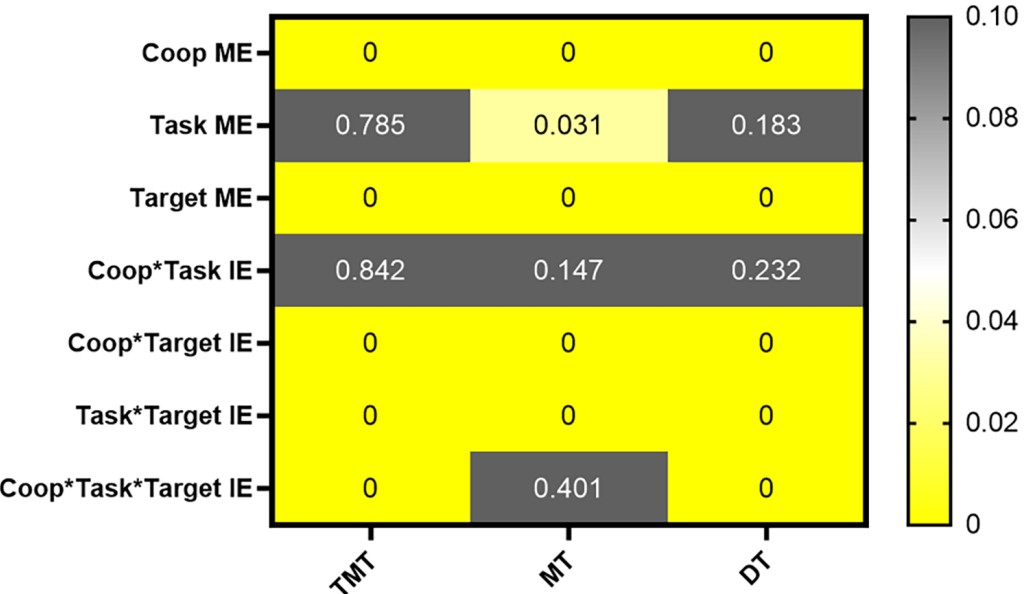

**Figure 3 The results of the analysis of holistic and localized indicators.** Coop ME represented the main effect of cooperative intention; Task ME represented the main effect of task characteristic; Target ME represented the main effect of the target; Coop*Task IE represented the interaction between cooperative intention and task characteristic; Coop*Target IE represented the interaction between cooperative intention and the target; Task*Target IE represented the interaction between task characteristic and the target; Coop*Task*Target represented the third-order interaction of cooperative intention, task characteristic, and the target. Yellow portions in the figure indicated significant differences, while gray portions indicated no significant differences.

key dwell time served as indicators of the message sender's sensorimotor communication performance. As the questionnaire strategy indicated that participants conveyed messages through target key dwell time, the subsequent analysis primarily focused on presenting the results related to target key dwell time. Detailed results for motion time and total motion time were provided in the Supplemental Information.

### Dwell time of target keys

A 2 (task characteristic: distance, orientation) × 2 (cooperative intention: Coop, No-coop) × 4 (target: T1, T2, T3, T4) repeated-measures ANOVA was conducted on the dwell time of the target key, and the results were displayed in Fig. 4. The analysis revealed the following significant effects and interactions: The main effect of cooperative intention was significant, $F_{(1, 64)} = 164.77$, $p < 0.001$, partial $\eta^2 = 0.72$. The main effect of target was significant, $F_{(1.56, 100.05)} = 59.19$, $p < 0.001$, partial $\eta^2 = 0.48$. The interaction of cooperative intention and target was significant, $F_{(1.56, 100.06)} = 59.58$, $p < 0.001$, partial $\eta^2 = 0.48$. The interaction of task characteristic and target was significant, $F_{(2.09, 133.42)} = 10.10$, $p < 0.001$, partial $\eta^2 = 0.14$. The triple interaction of task characteristic, cooperative intention, and the target was significant, $F_{(2.07, 132.61)} = 11.08$, $p < 0.001$, partial $\eta^2 = 0.15$. Further analysis revealed specific patterns: Target key dwell times for T1, T2, and T3 were greater than for T4 ($ps < 0.001$) under the distance task without cooperative intention. T1 target key dwell

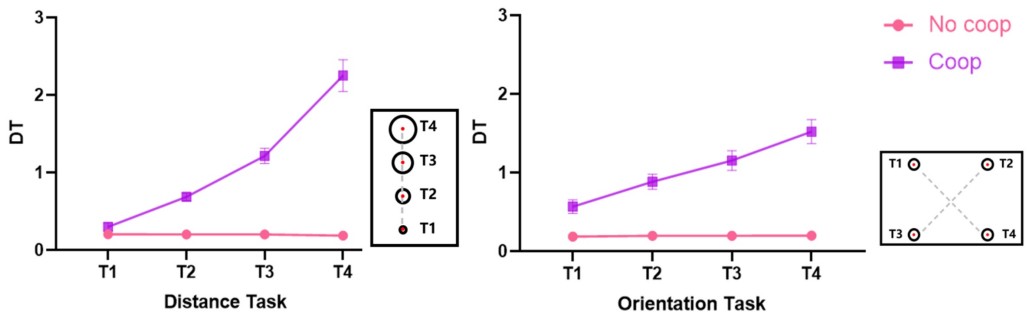

**Figure 4 Dwell time of target keys under different conditions.** Note: The straight lines in all graphs represent standard errors.

time was smaller than T2, T3, and T4 ($ps < 0.001$) under the orientation task without cooperative intention. Target key dwell time for T1, T2, T3, and T4 increased sequentially ($ps < 0.001$) under the distance task with cooperative intention. Target key dwell time for T1, T2, T3, and T4 showed a trend of sequential increase under the orientation task with cooperative intention. But there was no significant difference between T3 and T4 ($p > 0.05$), while the other two-by-two differences were significant ($ps < 0.05$) under the orientation task with cooperative intention. Comparisons between different conditions also yielded significant findings: The target key dwell time of T1 under the distance task without cooperative intention was greater than T1 under the orientation task without cooperative intention ($p < 0.001$). The target key dwell time of T4 under the distance task without cooperative intention was smaller than T4 under the orientation task without cooperative intention ($p = 0.004$). The target key dwell time of T1 under the distance task with cooperative intention was smaller than T1 under the orientation task with cooperative intention ($p = 0.006$). The target key dwell time of T4 under the distance task with cooperative intention was greater than T4 under the orientation task with cooperative intention ($p < 0.001$), while the remaining differences between experimental conditions were not significant ($ps > 0.05$). The results of the Bayesian repeated-measures ANOVA were generally consistent with these findings.

### Variability of target key dwell time

To thoroughly investigate the sensorimotor communication performance of message senders, this study further calculated the variability of target key DT ($SD_{DT}$) under different conditions and analyzed it using a repeated-measures ANOVA with a 2 (task characteristic: distance, orientation) × 2 (cooperation intention: Coop, No-coop) × 4 (target: T1, T2, T3, T4) design. The results revealed: A significant main effect of cooperative intention, $F_{(1, 64)} = 195.92$, $p < 0.001$, partial $\eta^2 = 0.75$. A significant main effect of the target, $F_{(2.24, 143.18)} = 40.00$, $p < 0.001$, partial $\eta^2 = 0.39$. A significant interaction between cooperative intention and target, $F_{(2.25, 144.18)} = 40.16$, $p < 0.001$, partial $\eta^2 = 0.39$. The interaction of task characteristic and target was significant, $F_{(2.05, 130.92)} = 3.11$, $p = 0.047$, partial $\eta^2 = 0.05$. The triple interaction of task characteristic, cooperative intention, and target was significant, $F_{(2.01, 128.53)} = 3.43$, $p = 0.04$, partial $\eta^2 = 0.05$. Subsequent simple

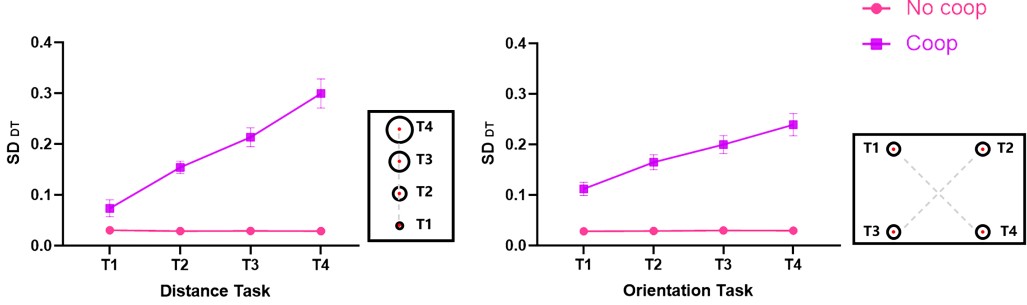

**Figure 5** Variability of target key dwell time under different conditions.

effect analyses indicated: that $SD_{DT}$ for T1, T2, T3, and T4 was not significant ($ps > 0.05$) for the distance task without cooperative intention and the orientation task without cooperative intention. $SD_{DT}$ for T1, T2, T3, and T4 increased sequentially ($ps < 0.05$) for the distance task with cooperative intention. $SD_{DT}$ for T1 was smaller than T2, T3, and T4 for the orientation task with cooperative intention, and T2's $SD_{DT}$ was smaller than T4 ($ps < 0.05$), as shown in Fig. 5. However, the Bayesian repeated-measures ANOVA did not find an interaction between task characteristic and target, and a triple interaction between task characteristic, cooperative intention, and target, and the rest of the findings were consistent with the above results.

Combining Figs. 4 and 5, it was observed that the longer the dwell time of the target key, the greater the variability observed in both distance and orientation tasks with cooperative intention.

*Quality of the message communication for target key dwell time*

The $SNR_{DT}$ of target key dwell time was analyzed by a 2 (task characteristic: distance, orientation) × 2 (cooperation intention: Coop, No-coop) repeated-measures ANOVA. The results indicated: A significant main effect of cooperative intention, $F_{(1, 64)} = 90.41$, $p < 0.001$, partial $\eta^2 = 0.59$. A significant main effect of task characteristic, $F_{(1, 64)} = 11.89$, $p = 0.001$, partial $\eta^2 = 0.16$. A significant interaction between cooperative intention and task characteristic, $F_{(1, 64)} = 23.39$, $p < 0.001$, partial $\eta^2 = 0.27$. Subsequent simple effects analyses revealed that: $SNR_{DT}$ was greater under the distance task with cooperative intention than without cooperative intention ($p < 0.001$). $SNR_{DT}$ was greater under the orientation task with cooperative intention than without cooperative intention ($p < 0.001$). For the distance task without cooperative intention, $SNR_{DT}$ was smaller than for the orientation task without cooperative intention ($p < 0.001$). $SNR_{DT}$ under the distance task with cooperative intention was greater than under the orientation task with cooperative intention ($p < 0.001$), as depicted in Fig. 6. The results of the Bayesian repeated-measures ANOVA were in perfect agreement with these findings.

## Movement trajectory

A repeated-measures ANOVA with a 2 (task characteristic: distance, orientation) × 2 (cooperative intention: Coop, No-coop) × 4 (target: T1, T2, T3, T4) design was conducted
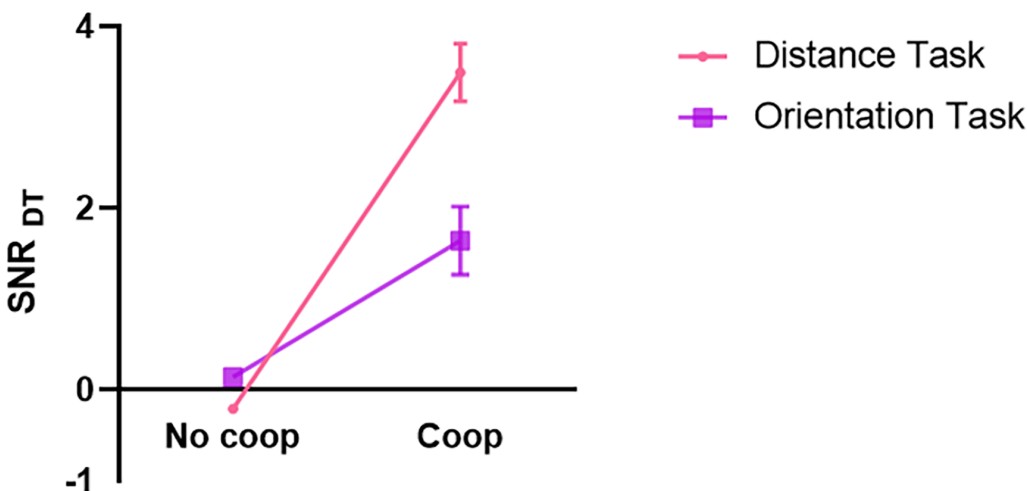

**Figure 6  Quality of signal exchange for different conditions of target key dwell time.**

on the maximum motion height ($MAX_{MH}$) from starting key press to target key lift. The results, as presented in Fig. 7, revealed the following: A significant main effect of cooperative intention, $F_{(1, 63)} = 13.50$, $p < 0.001$, partial $\eta^2 = 0.18$. A significant main effect of target, $F_{(2.69, 169.17)} = 163.07$, $p < 0.001$, partial $\eta^2 = 0.721$. A significant interaction between cooperative intention and target, $F_{(2.52, 158.54)} = 5.72$, $p = 0.002$, partial $\eta^2 = 0.08$. A significant interaction between task characteristic and target, $F_{(2.52, 158.54)} = 5.72$, $p = 0.002$, partial $\eta^2 = 0.71$. A significant triple interaction between task characteristic, cooperative intention, and target, $F_{(2.93, 184.26)} = 3.57$, $p = 0.016$, partial $\eta^2 = 0.05$. Subsequent simple effects analyses revealed: that $MAX_{MH}$ for T1, T2, T3, and T4 all sequentially increased (*ps* < 0.05) under both the distance task with and without cooperative intention. $MAX_{MH}$ for T1, T2, and T3 with cooperative intention was greater (*ps* < 0.05) than without cooperative intention. Under the orientation task, both with and without cooperative intention, the $MAX_{MH}$ of T3 was smaller than T1, T2, and T4 (*ps* < 0.05). The $MAX_{MH}$ of T1 was smaller than T4 with cooperative intention (*p* < 0.001), and the $MAX_{MH}$ of T2, T3, and T4 was larger with cooperative intention than without cooperative intention (*ps* < 0.05). The results of the Bayesian-based analysis largely corroborated these findings.

## Questionnaire

Through the organization of the questionnaire, 76.92% of the participants (50 persons) under distance task with cooperative intention extended their dwell time in proportion to the spatial distance information of the target key, drawing upon their previous sensory-motor experiences to establish a space–time mapping relationship between the task's spatial distance and their motion characteristic (target key dwell time). This resulted in a sequential increase in the target key dwell time for T1, T2, T3, and T4.

However, in the orientation task with cooperative intention, 47.69% of the participants (31 individuals) connected the four target locations in the order of left-up, right-up, left-down, and right-down according to embodied simulation and verbal metaphors. They increased the target key dwell time of T1, T2, T3, and T4 sequentially to establish the

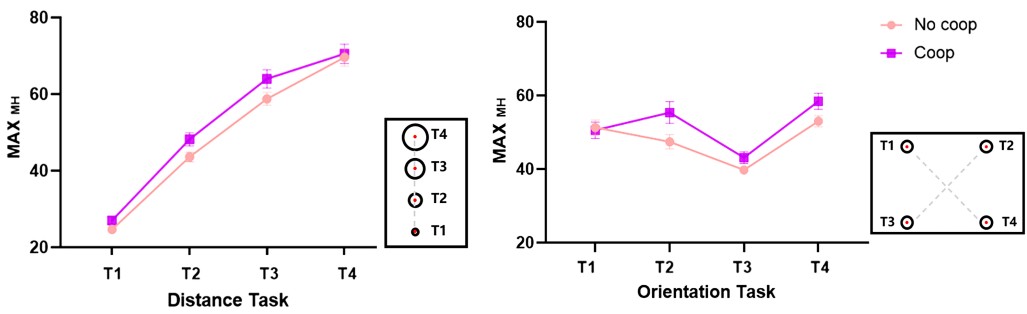

**Figure 7** **Maximum movement height for different conditions MAX$_{MH}$.**

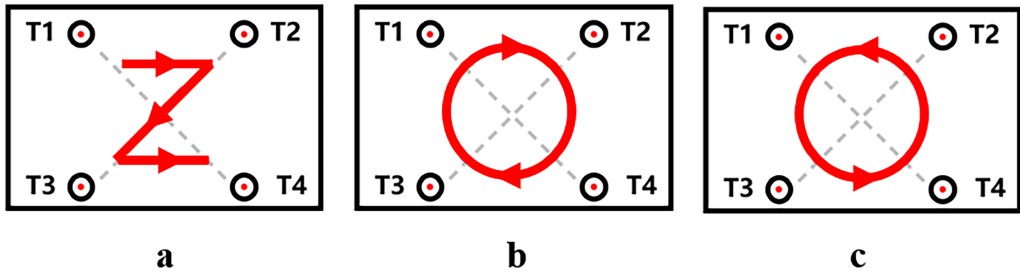

**Figure 8** **(A–C) Sensorimotor communication strategies of message senders under orientation task with cooperative intention.**

space–time mapping relationship. This strategy was defined as Strategy 1 (as shown in Fig. 8A). Additionally, 15.38% of the participants (10 individuals) employed a strategy where they established space–time mapping in clockwise order, connecting the four target positions in clockwise order and increasing the target key dwell time in turn. This strategy was labeled as Strategy 2 (Fig. 8B). Meanwhile, 9.23% of the participants (6 individuals) employed a counterclockwise order strategy, connecting the four target positions in counterclockwise order and sequentially increasing the target key dwell time. This strategy was defined as Strategy 3 (Fig. 8C). The remaining participants (23.08%, 15 individuals) used other strategies. Through K-center clustering analysis of target key dwell time SNR$_{DT}$, it was found that the index could be divided into four categories, corresponding to 12, 26, 16, and 11 cases, respectively. The corresponding clustering centers were 5.48, 2.98, 0.06, and −3.40, respectively, and the differences between the four categories were statistically significant ($F_{(3,61)} = 222.41$, $p < 0.001$). Combining the questionnaire results with the cluster analysis, it was observed that 87.10% of the participants who chose Strategy 1 in the questionnaire were clustered into categories 1 and 2.

## DISCUSSION

Building upon prior research, this study devised two asymmetric joint action tasks characterized by distinct spatial characteristics. It aimed to investigate the factors that drive sensorimotor communication in message senders by comparing different conditions. The

findings revealed the following insights. (1) Compared to conditions without cooperative intention, participants with cooperative intention exhibited significant increases in target key dwell time, motion time, total motion time, and maximum motion height. However, sensorimotor communication was primarily demonstrated through enhancements in target key dwell time. (2) In the distance task without cooperative intention, the dwell time of T4 is smaller than T1, T2, T3, and in the orientation task without cooperative intention, the dwell time of T1 is smaller than T2, T3, T4. Regardless of whether the distance task or orientation task was completed, there were no differences in the variability of dwell times of the four target keys without cooperative intention. Regardless of whether distance tasks or orientation tasks under cooperative intention, however, the dwell time of the target keys and their variability for T1, T2, T3, and T4 displayed a sequential increasing trend. In essence, a longer dwell time for the target key was associated with greater variability. (3) The quality of message communication related to target key dwell time and total motion time was superior with cooperative intention compared to conditions without cooperative intention in both distance and orientation tasks. Notably, the results were significantly more pronounced in the distance task with cooperative intention than in the orientation task with cooperative intention. (4) In the distance task with cooperative intention, nearly 80.00% of message senders established a space–time mapping based on sensorimotor experiences, characterized by "near-small, far-large". Conversely, in the orientation task with cooperative intention, nearly 50.00% of the message senders extended the dwell time of the target key in the order of "left-up, right-up, left-down, right-down".

## Sensorimotor communication for message senders with cooperative intention conditions

Prior research has shown that sensorimotor communication is widely present in cooperation (*Vesper & Sevdalis, 2020*). This aligns with the current study's discovery of significant disparities in the temporal characteristics (target key dwell time, motion time, and total motion time) and trajectory characteristics (maximum motion height) of message senders' actions when cooperative intention is present compared to when it is absent. However, it is essential to note that the dissimilarity in motion induced by cooperative intention does not necessarily equate to sensorimotor communication. For instance, the current study did not identify a third-order interaction among cooperative intention, task characteristics, and the target in terms of motion time. However, this interaction was observed in relation to the target key dwell time, total motion time, and maximum motion height of the target key. This suggests that sensorimotor communication by message senders may be reflected in these three motion characteristics.

In this study, the condition of cooperative intention was designed as a pseudocooperative task. It was explicitly conveyed to the message senders that the message receivers could hear their key presses but could not observe their motions. Notably, the message senders were unable to convey messages to their partners by altering the maximum motion height. The results indicated that message senders with cooperative intention exhibited higher maximum motion height compared to those without cooperative intention in both the distance and orientation tasks. However, the patterns of change in the four target locations

with and without cooperative intentions were very similar. These differences may therefore stem from variations in the instrumental actions associated with the target key as well as generalized effects arising from cooperative intention. Consequently, sensorimotor communication by message senders is primarily expressed through the dwell time and total motion time of the target key.

However, it is worth noting that total motion time might not be the most accurate indicator of sensorimotor communication. Total motion time is a holistic metric that encompasses multiple phases of motion and is influenced by various factors. An examination revealed that when the proportion of target key dwell time in total motion time was removed, the results closely resembled the patterns observed in maximum motion height. This implies that sensorimotor communication within total motion time is mainly reflected in the target key dwell time. Additionally, the findings from the strategy questionnaire further corroborated the finding that message senders primarily rely on target key dwell times for communication.

In summary, it is evident that sensorimotor communication is indeed contingent on cooperative intention, but it is not evident across all phases of motion. Previous studies, such as those conducted by *Vesper & Richardson (2014)* and *Laroche et al. (2022)*, have primarily explored the dissociation of specific motion phases induced by sensorimotor communication. In contrast, the present study offers a comprehensive evaluation of sensorimotor communication performance by message senders, encompassing local and holistic as well as temporal and trajectory perspectives. Consequently, the relationship between sensorimotor communication and cooperative intention is more robust and dependable.

## Sensorimotor communication performance of message senders in different task characteristics

### Sensorimotor communication performance of message senders in a distance task with cooperative intention

The current study revealed that in a distance task with cooperative intention, message senders extended their target key dwell time proportionally to the spatial distance of the task target, in alignment with Hypothesis 2a. These findings were in line with prior research (*Vesper, Schmitz & Knoblich, 2017*; *Castellotti et al., 2022*; *Chen et al., 2021*). The theoretical framework of embodied cognition suggests that an individual's understanding of the world commences with bodily perception. The construction and comprehension of abstract concepts rely on sensorimotor experiences and involve an automated perceptual simulation process (*Wang et al., 2020*; *Ye, Zeng & Yang, 2019*; *Di Paolo, Cuffari & De Jaegher, 2018*; *Li, 2008*). When individuals process abstract concepts, their prior sensorimotor experiences are automatically activated, potentially influencing their current action performance. Therefore, in a distance task with cooperative intention, when message senders engaged with the task target, the distance information associated with the target triggered previous sensorimotor experiences. This, in turn, prompted individuals to simulate their action performance, resulting in prolonged dwell time for the target key as the distance increased. Consequently, they effectively conveyed task target information to others.

Furthermore, the present study demonstrated that the variability in message senders' target key dwell time progressively increased from T1 to T4 in both distance and orientation tasks. This finding was consistent with prior research (*Castellotti et al., 2022*). Notably, individual differences in estimating shorter durations were significantly smaller than for longer durations (*Huang, 2022*).

### Performance of sensorimotor communication by message senders in an orientation task with cooperative intention

In the orientation task with cooperative intention, message senders extended the target key dwell time proportionally to the orientation sequence of target positions, left-up, right-up, left-down, and right-down, to effectively convey their message. This observation aligned with Hypothesis 2b. The questionnaire responses further indicated that 47.69% of the participants consciously established this space–time mapping relationship, providing support for the hypothesis. Furthermore, the variability in target key dwell time also increased as the dwell time was extended, which was consistent with previous research findings (*Castellotti et al., 2022*; *Huang, 2022*).

Previous studies in the field of the Space-Time Association of Response Codes (STARC) have identified mental timelines associated with the "left–right" and "up-down" orientations (*Casasanto & Bottini, 2014*; *He et al., 2021*). However, these investigations primarily explored the space–time mapping relationship from a one-dimensional spatial perspective. The current study extended this understanding by providing empirical evidence for a two-dimensional STARC effect. Specifically, individuals perceived time as passing least quickly in the left-up position, followed by the right-up and the left-down, with the longest duration in the right-down position. Prior research has also noted that individuals exhibit a more pronounced STARC effect in the horizontal direction than in the vertical direction. In this context, the mental timeline effect associated with the horizontal direction tended to dominate between the two mental timelines (*Yang & Sun, 2016*). Researchers have observed that individuals typically associate the left-up position with shorter durations and the right-down position with longer durations (*Sun et al., 2022*).

Comparison of sensorimotor communication for message senders with different task characteristics. Previous research has indicated that various factors, such as gender and emotional state (*Zhao et al., 2020*) as well as role (*Candidi et al., 2015*), influence the dynamics of sensorimotor communication among interacting parties. The current study extended this body of research by revealing that task characteristics also exerted an impact on individuals' sensorimotor communication. Specifically, the study showed that target key dwell time, exhibited by message senders during both distance and orientation tasks with cooperative intention progressively increased from T1 to T4. However, a subtle distinction emerged between these two task types. Notably, for T1, the target key dwell time was significantly shorter during the distance task than during the orientation task. Conversely, for T4, the opposite trend was observed. These differences underscore the influence of task characteristics on sensorimotor communication.

Furthermore, the quality of the message communication for target key dwell time was higher in the distance task with cooperative intention compared to the orientation task with

cooperative intention. Specifically, the distance-time mapping relationship established by individuals based on their sensorimotor experiences appeared to be relatively clear during the distance task with cooperative intention and was characterized by more consistent and proportionally varying temporal responses across different target distances. In contrast, during the orientation task with cooperative intention, although an orientation-time mapping relationship was evident and exhibited a gradual increase from left-up, right-up, left-down, to right-down, it lacked a specific representation of different orientations, resulting in a less clear and proportionally varying temporal response.

## Strategies for sensorimotor communication by message senders in different tasks

The strategies employed by message senders with cooperative intention differed depending on the task at hand. In the distance task with cooperative intention, 76.92% of message senders prioritized conveying target information through sensorimotor experience. This manifested as a sequential increase in the target key dwell time for T1, T2, T3, and T4. Conversely, in the orientation task with cooperative intention, message senders utilized a more varied set of strategies to convey information. Three additional strategies emerged in this task: associating the orientation of the four target keys with dwell time in the sequence of left-up, right-up, left-down, right-down, following either a clockwise or counterclockwise order, and increasing the target key dwell time accordingly. Among these strategies, the most frequently used strategy was the first, which combined sensorimotor experience and verbal metaphors, accounting for approximately 50%. This indicated that at the group level when the task allowed for it, message senders typically established space–time mappings rooted in their sensorimotor experiences. Importantly, the formation of these space–time mapping relationships by message senders was not predetermined with message receivers but emerged spontaneously within group dynamics (*Grasso et al., 2022*). This finding underscored the substantial influence of previous sensorimotor experiences on group behavior (*Zhang et al., 2022*).

## Limitations and Outlook

This study successfully controlled for the objective difficulty of different target keys according to *Fitts*' law (*1954*). However, it is worth noting that specific performance variations emerged in motion time between the four target positions in both the distance and orientation tasks without cooperative intention. These differences might be attributed to variations in the ease of pressing the actual target keys. Consequently, future research should consider not only the objective difficulty of key presses but also the influence of individual physical limitations. In addition, the present study only examined the space–time mapping relationship of sensorimotor communication in Mandarin-speaking participants. Culture may have an impact on the space–time mapping relationship. Future studies could also examine the space–time mapping relationship of sensorimotor communication across cultures. Furthermore, the neural underpinnings of sensorimotor communication remain largely unexplored. Future investigations could utilize advanced techniques such as functional nuclear magnetic resonance (fMRI) to pinpoint the specific brain regions

or networks involved in sensorimotor communication. Additionally, employing methods such as event-related potential (ERP) and functional near-infrared spectroscopy (fNIRS) could shed light on the interbrain mechanisms underlying sensorimotor communication within real communication contexts. These advancements will contribute to a more comprehensive understanding of the phenomenon.

## CONCLUSIONS

(1) Compared to situations without cooperative intention, when cooperative intention is present, message senders tend to exaggerate certain kinematic characteristics during various motion phases as a means to facilitate sensorimotor communication. Notably, the primary aspect through which sensorimotor communication is expressed is the dwell time of the target key.

(2) Sensorimotor communication primarily relies on the mapping relationship between the task target and the message sender's motion characteristics, as exemplified by: In the distance task with cooperative intention, message senders predominantly utilize the sensorimotor experience of "near-small, far-large" to convey task information. Conversely, in the orientation task with cooperative intention, message senders primarily utilize a combination of "left-up, right-up, left-down, right-down" sensorimotor experiences along with verbal metaphors to convey task information.

## ACKNOWLEDGEMENTS

We would like to thank all the participants who took part in the experiment.

### Funding

This work was supported by the National Nature Science Foundation of China (No. 32100878) and the Tianjin Normal University Doctoral Foundation (No. 52WW2104). The funders had no role in study design, data collection and analysis, decision to publish, or preparation of the manuscript.

### Grant Disclosures

The following grant information was disclosed by the authors:
National Nature Science Foundation of China: 32100878.
Tianjin Normal University Doctoral Foundation: 52WW2104.

### Competing Interests

The authors declare there are no competing interests.

### Author Contributions

- Ke Zhang performed the experiments, analyzed the data, prepared figures and/or tables, authored or reviewed drafts of the article, and approved the final draft.

- Xin Tong performed the experiments, prepared figures and/or tables, and approved the final draft.
- Shaofeng Yang analyzed the data, prepared figures and/or tables, and approved the final draft.
- Ying Hu performed the experiments, prepared figures and/or tables, and approved the final draft.
- Qihan Zhang conceived and designed the experiments, performed the experiments, analyzed the data, authored or reviewed drafts of the article, and approved the final draft.
- Xuejun Bai conceived and designed the experiments, authored or reviewed drafts of the article, and approved the final draft.

## Human Ethics

The following information was supplied relating to ethical approvals (*i.e.*, approving body and any reference numbers):

Tianjin Normal University granted ethical approval to carry out the study within its facilities (Ethical aoolication Ref: 2021030809).

## Data Availability

The final dataset used for the analysis is available in the Supplemental File.

## Supplemental Information

Supplemental information for this article can be found online at http://dx.doi.org/10.7717/peerj.16764#supplemental-information.

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
