# Peer review of "Space-time mapping relationships in sensorimotor communication during asymmetric joint action"

_PeerJ, doi:10.7717/peerj.16764_

## Round 0.1 · original submission · Major Revisions

All the reviewers raise major concerns.

Reviewer 1 ·

Basic reporting

The article titled "Space-Time Mapping Relationships in Sensorimotor Communication during Asymmetric Joint Action" investigates the performance of message senders in sensorimotor communication, focusing on motion time and motion amplitude. However, the article requires significant revision in various areas, as outlined below.

Experimental design

1、 The criteria for determining the sample size were not disclosed. Please provide relevant information to help readers understand how the sample size was determined.
2、 In this study, information transfer primarily relied on motion time. Did the participants' prior experience with motion and time control influence the experiment's results? If so, was this controlled for in the study?
3、 Even if Participant B was virtual in this study, research has demonstrated that the gender of a companion can impact joint action. Were steps taken to control for the influence of the companion's gender on sensorimotor communication during the experiment?

Validity of the findings

The article assesses sensorimotor communication performance across different time perspective trajectories, and through subjective questionnaires, offering a comprehensive description of the analytical methods. However, some issues remain.
4、 Does it contradict the research hypothesis 2b that participants' reported strategies in the orientation task with cooperative intention included clockwise and counterclockwise strategies?
5、 According to the experimental settings, pressing the start key will also produce a sound. Will the participants transmit information through the sound of the start key?

Additional comments

Numerous statements in the article require revision.
6、 Descriptions of the customized keyboard are scattered throughout different paragraphs; therefore, it is recommended to consolidate them in a single location.
7、 Keystroke difficulty is calculated using Fitts' law, and it is advisable to provide the formula for clarity.
8、 Lines 432-435, 541, 550 the results of the report were not supported by the result of questionnaire in this article.
9、 Lines 98-101 contain convoluted sentences that are challenging to comprehend. A rewrite is suggested.
10、 In line 140, a specific theory requires the addition of "the."
11、 At lines 233 and 400, please review the format of singular and plural nouns.
12、 Lines 279-281 exhibit inconsistent verb tenses, while line 307 contains incorrect verb formatting.
13、 The name of the statistical method is wrong. "Paired t tests" should be written as "paired t-tests."
14、 In Figure 4, the vertical coordinate is missing.
15、 Lines 364-367 feature overly long sentences, and splitting them into two sentences is recommended. Please check for similar issues in other parts of the text.
16、 In lines 375-376, there are redundant spaces. Please verify if this issue occurs elsewhere in the document.
17、 In the presentation of experimental data, such as in Figure 7, it is unclear whether the dotted line represents standard deviation or standard error. Please provide a detailed explanation. Also, clarify why standard deviation/error is shown for some conditions but not others.
18、 Line 526 contains an incorrectly formatted literature citation.
19、 The references contain an excessive use of "°"; please make the necessary revisions.

·

Basic reporting

The study "Space-time mapping relations in sensorimotor communication during asymmetric joint action" investigates the cognitive mechanisms underlying sensorimotor communication. While the study holds research value, extensive revisions are required before publication.

Experimental design

The methodology is detailed and precise, but the study needs improvements to enhance its precision and credibility. The following issues should be addressed:
1. Explain why a fixed order was used instead of sequential counterbalancing between tasks with and without cooperative intentions.
2. Suggest including a flowchart for the no cooperative intention task in Figure 2 to ensure completeness.
3. Separate the presentation of the experimental apparatus and the experimental setup for clarity.
4. how to prove that the sensorimotor communication of participants is effective in the experimental tasks?

Validity of the findings

1. Why the degrees of freedom for the F-values are not integers?
2. please supplement the data collation of the motion capture data in this study.

Additional comments

1. The abstract highlights the importance of cooperative intention as a prerequisite for sensorimotor communication, but the example in line 57 suggests sensorimotor communication can occur between opponents. There is a contradiction.
2. In line 125, "message senders with cooperative intention" might be misunderstood as referring to between-group design. It means " message senders in the condition with cooperative intention".
3. Participants in Sun et al.(2022) were Mandarin-speaking, and whether the study applied to this group.
4. In line 209, eliminate the repeated use of "and" in "T1, T2, T3, and T4 and the starting key" for clarity.
5. Standardize using singular and plural for "target cue" in lines 221 and 224.
6. Maintain consistency between "individual" and "participant".
7. "Prior research has also noted that individuals exhibit a more pronounced STARC effect in the horizontal direction than in the vertical direction" lacks a reference.
8. In line 609, use the full names of ERP and fNIRS the first time the abbreviations appear.
9. Revise the conclusions section to distinguish between results and conclusions.

Reviewer 3 ·

Basic reporting

The study employed metrics such as motion time and amplitude to examine how individuals conveyed information through their motions with Cooperative intention. I recommend that the authors undertake more comprehensive revisions in the literature review and discussion.

Experimental design

While the experimental design was generally sound, there is room for improvement. Here are specific suggestions:
1. The authors should explicitly detail the method used to calculate the sample size in the subject selection section. Additionally, when using MorePower software for sample size calculation, the report should include various parameters, such as whether it is a one-tailed test and the effect sizes.
2. Lines 175, 178-179, the notation "Mage=20.06 years, SDage=2.80 years" and "Marm=68.22 cm, SDarm=4.87 cm; Mheight=169.66 cm, SDheight=9.42 cm" should be adequately subscripted.

Validity of the findings

The authors have shared raw data and appropriately applied exclusion criteria. The ANOVA was executed correctly, and specific effect sizes were reported.

Additional comments

1. The introduction and discussion refer to "sensorimotor experience" without defining the term. It is recommended to provide a concise definition of "sensorimotor experience".
2. The introduction mentions that athletes use deceptive body movements to disrupt opponents' motion prediction processes, implying sensorimotor communication in competitive situations. However, the discussion suggests that "sensorimotor communication must be predicated on cooperative intentions between individuals." This contradiction needs revision.
3. what’s the theoretical and practical significance of the cognitive mechanism in sensorimotor communication.
4. lines 41-43, in the study of Schmitz et al (2018), did all participants use this pattern to convey the task information?
5. Line 64 contains a grammatical error regarding singular/plural agreement.
6. Line 76 contains unclear language. "for sensorimotor to enable asymmetric joint action" should be replaced with "for sensorimotor communication".
7. Lines 92-94 use embodied cognition theory to explain the two studies, but only the second. Clarify this inconsistency.
8. In line 159, consider providing specific examples for the Chinese terms "morning" and "afternoon".
9. Line 163 contains a grammatical error and inconsistent verb tenses.
10. In line 257, "changes in the target key arrangement and the use of the orientation keyboard" is repetitive and should be rephrased.
11. The Discussion states that "sensorimotor communication is mainly reflected in the dwell time of the target key and total motion time.". But the Conclusion says, "sensorimotor communication is mainly reflected in the dwell time of the target key." It's advisable to maintain consistency to avoid confusion.
12. In lines 498 and 499, "the patterns of change across different target locations were remarkably similar", please specify which levels these patterns refer to.
13. Given that the study was conducted in China and the potential impact of Chinese writing and reading habits on the findings, it is recommended that possible cultural differences be recognized.

---

## Round 0.2 · Minor Revisions

There are still some minor comments that should be addressed.

Reviewer 1 ·

Basic reporting

I have checked all your responses and revisions. The revised manuscript has improved quite a lot. But I found many places that need to be redone.

Experimental design

1、 There is a discrepancy between the description in the paragraph and Fig. 2a, which is the distance task without cooperative intention, at line 258.

Validity of the findings

2、 Statistical indicators in lines 178, 181, 380, and 382 should be italicized.

Additional comments

3、 It is advisable to provide more detailed information regarding the changes in motion parameters for the Coop relative to the No-coop intention in the abstract of results (1).
4、 Bold formatting is recommended for lines 187, 531-532. Additionally, line 285 needs to be checked for errors in the formatting of the Keystroke Response.
5、 In line 218, "a participant" should be corrected to "a participant".
6、 “Regardless of whether distance tasks or orientation tasks were completed, there were no differences in the dwell times of the four target keys and their variability without cooperative intention.” Please verify line 477 as the Results report a difference.
7、 A period is missing at line 549.
8、 Lines 602、634, for consistency, it is recommended to harmonize the use of sensory-motor experiences, as most of the text uses "sensorimotor experiences".
9、 Line 621 should include the full name of fMRI.
10、 The use of “sensorimotor communication along with verbal metaphors” should be checked for problematic issues at line 637.

·

Basic reporting

The author has made careful revisions,the quality of the paper has been greatly improved. I am satisfied with the author's reply, and I recommend accepting this article.

Experimental design

no comment

Validity of the findings

no comment

Additional comments

There are also some minor errors in the language, such as singular and plural usage, a/an usage, etc., please check carefully.

Reviewer 3 ·

Basic reporting

The quality of the revised draft has improved considerably. My question was well answered.

Experimental design

no comment

Validity of the findings

no comment

Additional comments

1. line 27-31, It is suggested that the results of the no-coop conditions should also be reported.
2. line 32-38, It is suggested that the relationship between sensorimotor experience, verbal metaphor and space-time mapping in sensorimotor communication should be pointed out in the conclusion.
3. line 97, “solitary task” is less accurate, maybe "the tasks without cooperation" is more suitable.
4. line 187, "experimental design" needs to be bolded.
5. line 202, The company name is misspelled, should be: Beijing Nokov Science & Technology.

---

## Round 0.3 · accepted · Accept

This manuscript can be accepted now.

Reviewer 1 ·

Basic reporting

I have carefully reviewed the manuscript, and I am pleased to note that the experimental design, data analysis, and discussion in this article demonstrate a high level of scientific rigor after revisions of the authors. The authors have successfully met the publication requirements, and the overall quality of the research is commendable. I recommend acceptance for publication.

Experimental design

No comments.

Validity of the findings

No comments.

Additional comments

No comments.

Reviewer 3 ·

Basic reporting

no comment

Experimental design

no comment

Validity of the findings

no comment

Additional comments

no comment